# Disordered RNA chaperones can enhance nucleic acid folding via local charge screening

Erik D. Holmstrom [1,4], Zhaowei Liu [1], Daniel Nettels[1], Robert B. Best [2] & Benjamin Schuler [1,3]

RNA chaperones are proteins that aid in the folding of nucleic acids, but remarkably, many of these proteins are intrinsically disordered. How can these proteins function without a well-defined three-dimensional structure? Here, we address this question by studying the hepatitis C virus core protein, a chaperone that promotes viral genome dimerization. Using single-molecule fluorescence spectroscopy, we find that this positively charged disordered protein facilitates the formation of compact nucleic acid conformations by acting as a flexible macromolecular counterion that locally screens repulsive electrostatic interactions with an efficiency equivalent to molar salt concentrations. The resulting compaction can bias unfolded nucleic acids towards folding, resulting in faster folding kinetics. This potentially widespread mechanism is supported by molecular simulations that rationalize the experimental findings by describing the chaperone as an unstructured polyelectrolyte.

[1] Department of Biochemistry, University of Zurich, Winterthurerstrasse 190, 8057 Zurich, Switzerland. [2] Laboratory of Chemical Physics, National Institute of Diabetes and Digestive and Kidney Diseases, National Institutes of Health, Bethesda, MD 20892-0520, USA. [3] Department of Physics, University of Zurich, Winterthurerstrasse 190, 8057 Zurich, Switzerland. [4] Present address: Department of Molecular Biosciences, University of Kansas, Lawrence, KS 66045-7566, USA. Correspondence and requests for materials should be addressed to E.D.H. (email: erik.d.holmstrom@ku.edu) or to R.B.B. (email: robert.best2@nih.gov) or to B.S. (email: schuler@bioc.uzh.ch)

Ever since the discovery of transfer RNA and ribosomal RNA, the number of functions attributed to structured nucleic acids has continuously increased[1]. Due to the polyanionic nature of nucleic acids, formation of these structures is strongly influenced by electrostatics. Additionally, as in the case of proteins, non-native interactions can impede the molecular search for biologically active conformations and result in kinetic traps[2]. The folding problem for nucleic acids is exacerbated by their small set of monomeric building blocks, which form strong interactions with each other, making it difficult for a given sequence to specify a unique structure[3]. As a result, oligonucleotides often adopt long-lived non-native conformations with impaired function. RNA chaperones are a class of proteins that alleviate this problem by helping nucleic acids navigate their rugged energy landscape towards compact and thermodynamically stable conformations[3–11].

An important group of RNA chaperones are viral nucleocapsid proteins, which facilitate many nucleic-acid-dependent processes in viral life cycles[12–15]. Comparisons among these chaperones reveal very few structural similarities, but many of them are enriched in positively charged and polar residues, resulting in proteins that are known or predicted to be disordered[13,16,17]. Their chaperone activity is present both in vitro and in vivo[3,5,8,12,13], and the intrinsically disordered regions are both necessary and sufficient for this function[15]. These findings give rise to a question of fundamental importance: How do RNA chaperones function without a well-defined structure?

We address this question by studying NCD, the nucleocapsid domain of the hepatitis C virus core protein (Fig. 1a), using a variety of single-molecule fluorescence experiments. NCD is intrinsically disordered as a monomer[18,19] and has many important roles in the life cycle of this virus, which infects one in 50 people worldwide[20]. In addition to forming the nucleocapsid enclosing the viral RNA, NCD also acts as an RNA chaperone and facilitates many nucleic-acid conformational transitions[15], including dimerization of the viral genome (Supplementary Fig. 1) via a self-complementary hairpin[21]. To understand how intrinsically disordered RNA chaperones function, we study NCD's ability to alter the folding kinetics of a ubiquitous nucleic acid structural motif—a stem-loop hairpin (Fig. 1a). We note that the tendency for NCD to form nucleocapsid-like particles in the presence of nucleic acids[19,22] prevents investigation of this process at the high sample concentrations required for many ensemble techniques, such as circular dichroism or NMR spectroscopy. Therefore, we use an integrative combination of single-molecule Förster resonance energy transfer (FRET) spectroscopy and molecular simulations to elucidate the mechanism of chaperone-assisted folding. We find that NCD facilitates the folding of nucleic acids by acting as a macromolecular counterion that locally screens charge repulsion with extreme efficiency. Note that in this work, the terms RNA chaperone and chaperone are broadly used to refer to "proteins that aid in the folding of RNA" rather than the more restrictive definition[3] of "proteins that aid in the process of RNA folding by preventing misfolding or by resolving misfolded species", following the common classification of NCD as an RNA chaperone[15,23,24].

## Results

### NCD promotes folding via electrostatic interactions.

The capacity for NCD to chaperone hairpin formation is apparent from confocal single-molecule FRET experiments on freely diffusing nucleic acids designed to form a 7-bp stem-loop structure (Fig. 1a). These hairpin molecules were site-specifically labeled near the 3′ and 5′ ends with donor and acceptor dyes, respectively, allowing the formation of the stem-loop (i.e., folding) to be monitored via a change in the highly distance-dependent transfer efficiency. In the absence of NCD and at near-physiological concentrations of monovalent cations, the hairpin primarily occupies unfolded conformations, which result in a low mean transfer efficiency, $\langle E \rangle$. With increasing concentration of NCD (Fig. 1b, c), the hairpin more frequently occupies high-transfer-efficiency folded conformations. The low-nanomolar apparent dissociation constant, $^{app}K_d$, of this chaperone-induced transition (Fig. 1c) highlights the high affinity of NCD for the DNA hairpin. Similar results were observed for experiments with an RNA hairpin (Supplementary Fig. 2a). Furthermore, experiments on an oligonucleotide with identical length and a sequence that is unable to form a hairpin reveal that the chaperone also binds and compacts this folding-incompetent nucleic acid (Supplementary Fig. 2b).

Given the large and opposite structural charge[25] of NCD (+25/−3) and the DNA (−60) (Fig. 1a), their association is expected to be electrostatically driven. Indeed, the stability of the complex is extremely sensitive to the addition NaCl, with a strong concomitant increase in $^{app}K_d$ (Fig. 1c, d). This effect can be interpreted[26] in terms of the release of ~14 counterions when NCD and the DNA associate (Supplementary Fig. 3), a substantially larger number than previously reported for other nucleic acid-binding proteins[27,28]. Notably, the addition of physiological concentrations of $Mg^{2+}$ does not alter NCD's ability to promote hairpin formation (Supplementary Fig. 4a). In low-salt conditions, multiple chaperone molecules can bind a single hairpin (Supplementary Fig. 4b), presumably because of the charge imbalance between NCD and the hairpin. However, in the following sections, we focus on the one-to-one complex (see Methods section). To summarize, the strong electrostatic contribution to binding suggests that the interaction is generic for different kinds and sequences of nucleic acids and rationalizes our finding that NCD has the capacity to bind both DNA and RNA as well as folded and unfolded molecules.

### Chaperone binding accelerates folding.

To observe chaperone-assisted folding directly, we recorded fluorescence time traces of individual surface-immobilized 5′-3′ FRET-labeled hairpin molecules using confocal fluorescence spectroscopy. In the absence of NCD (Fig. 2a), the average dwell time in the low-transfer-efficiency unfolded (U) conformation, $\langle \tau_U \rangle = 60 \pm 20$ ms, is more than twice as long as the average dwell time in the high-transfer-efficiency folded (F) conformation, $\langle \tau_F \rangle = 26 \pm 8$ ms. The resulting transfer efficiency histograms resemble those from free-diffusion measurements under identical conditions (Fig. 1b; red), indicating that immobilization does not perturb the system. With 150 nM unlabeled NCD (Fig. 2b), a concentration where the hairpin is bound to chaperone for more than 95% of the time (Fig. 1c), the fluorescence time traces look very different from those in the absence of NCD: The transfer efficiency of the hairpin is almost always high because NCD favors the chaperone-bound folded conformation of the hairpin (FC). However, there are frequent short excursions to a second conformation with an average dwell time of $\langle \tau_{UC} \rangle = 3 \pm 1$ ms, which give rise to a small population at a transfer efficiency of ~0.5 (Fig. 2b, d). This small population is also visible in the free-diffusion measurements under identical conditions (Fig. 1b; cyan), and its transfer efficiency is close to that of the folding-incompetent hairpin bound to NCD (Supplementary Fig. 2b). These observations suggest that the chaperone-bound unfolded hairpin (UC) is transiently populated at equilibrium, and that it is more compact than the chaperone-free unfolded hairpin (U). The existence of the UC population was further confirmed by recurrence analysis of independent free-diffusion measurements[29] (Supplementary Fig. 5).

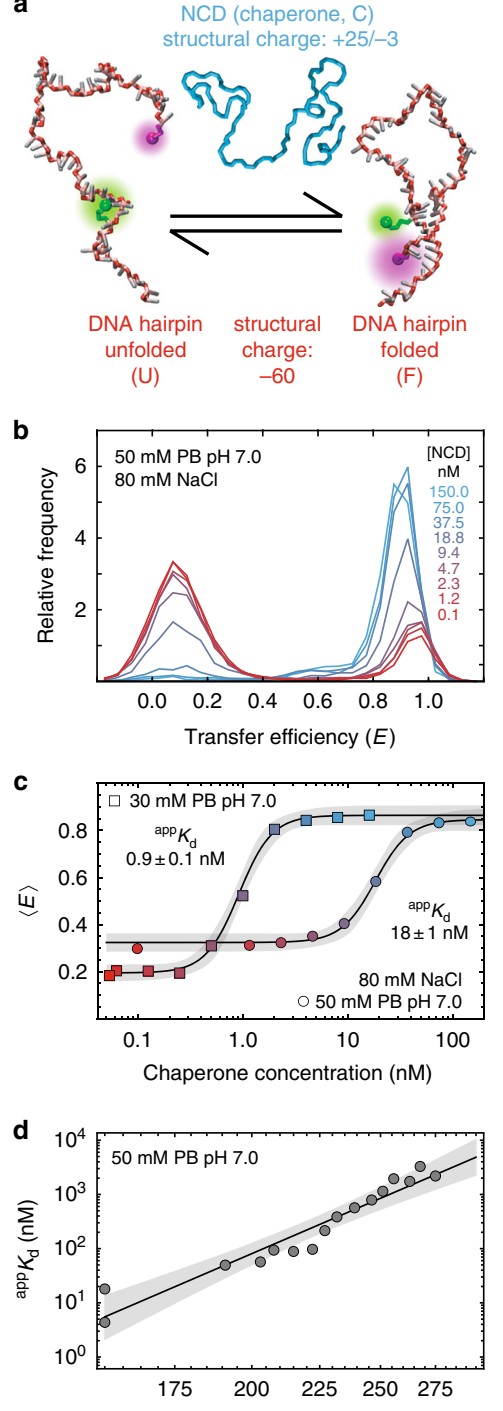

**Fig. 1** Nucleic-acid chaperone binds hairpin with high affinity. **a** Molecular representations of the folded and unfolded DNA hairpin with the phosphate, sugar, and base moieties colored in red, pink, and white, respectively; magenta and green spheres indicate the acceptor and donor dyes within the 5′-3′ FRET-labeled hairpin. The positively charged and intrinsically disordered nucleocapsid domain of the hepatitis C virus core protein (NCD, cyan) chaperones hairpin formation. **b** Transfer efficiency histograms from measurements of labeled hairpin at increasing concentrations of unlabeled chaperone under conditions with near-physiological concentrations of monovalent cations (~75 mM from phosphate buffer (PB) and 80 mM from NaCl). **c** Mean transfer efficiencies, $\langle E \rangle$, from **b** plotted against chaperone (NCD) concentration. Source data for mean values are provided as a Source Data file. Apparent dissociation constants, $^{app}K_d$, are determined from fits to a 2-state binding isotherm (see Methods section). **d** Double-logarithmic dependence of $^{app}K_d$ on sodium ion concentration with fit to a power-law of the form $f(x) = ax^n$ (solid line, 99% confidence band in gray), where $x$ is the concentration of sodium ions and $n$ is related to the number of counterions released when NCD binds to the hairpin[62]

roughly half of the time. Under these conditions (Fig. 2e), fluorescence time traces show single molecules toggling between the two distinct kinetic regimes (Fig. 2a, b) as the chaperone binds and dissociates on the timescale of seconds. The single-molecule experiments report on chaperone binding primarily via changes in folding/unfolding kinetics, as the transfer efficiencies of F and FC are very similar. Therefore, we analyzed the fluorescence time traces using a four-state hidden-Markov model (Fig. 2f) (see Methods section). Maximum-likelihood (MLH) analysis shows that NCD binds to the unfolded and folded hairpin with similar rate constants of ~$10^8\,M^{-1}\,s^{-1}$, but that it dissociates an order of magnitude more slowly from the folded than from the unfolded hairpin (Fig. 2f; gray). The resulting binding and dissociation rate constants, as well as the $K_d$ for NCD and the unfolded (U) hairpin were consistent with those observed for surface-immobilized folding-incompetent hairpins (Supplementary Fig. 6), where these rate constants can be measured independently, confirming both the accuracy of the MLH analysis and the applicability of the four-state model (Fig. 2f). In summary, our results suggest that NCD aids in nucleic acid folding by increasing the free energy of the unfolded relative to the folded state, which leads to a concomitant reduction in the folding free-energy barrier and shifts the equilibrium towards folded conformations. It is worth mentioning that this mechanism is not expected to specifically prevent misfolding.

**A disordered protein-nucleic acid complex.** What are the conformational properties of this protein-nucleic acid complex, and how do they explain the molecular mechanism by which NCD accelerates folding? There is increasing evidence that positively charged intrinsically disordered proteins can form high-affinity complexes with nucleic acids and remain disordered and highly dynamic in the bound state[30,31], including NCD in complex with viral RNA hairpins[19], suggesting that NCD performs its chaperoning function as a disordered protein. To test whether it indeed remains disordered and correspondingly dynamic after binding to the hairpin, we labeled a two-cysteine variant of NCD (S2C-T65C) with donor and acceptor dyes (Fig. 3a) and used nanosecond fluorescence correlation spectroscopy[32] to probe its reconfiguration dynamics in the presence and absence of the unlabeled hairpin and its folding-incompetent counterpart (Fig. 3b). In all cases, the resulting correlation functions reveal pronounced fluorescence fluctuations of donor and acceptor emission on the 50-ns timescale, as expected for the long-range

In both the absence (Fig. 2c) and presence (Fig. 2d) of NCD, a quantitative two-state kinetic analysis (see Methods section) of the dwell times yielded rate constants consistent with the donor-acceptor fluorescence cross-correlation functions. A comparison of the rate constants in the two distinct kinetic regimes reveals that hairpin folding is accelerated more than 20-fold when NCD is bound, with no significant change in the unfolding rate constant. Again, these results agree with a recurrence analysis of corresponding free-diffusion experiments (Supplementary Fig. 5), providing further support that immobilization does not alter the behavior of the hairpin or NCD.

The full kinetics of chaperone-assisted folding are most obvious at NCD concentrations near $^{app}K_d$, where NCD is bound

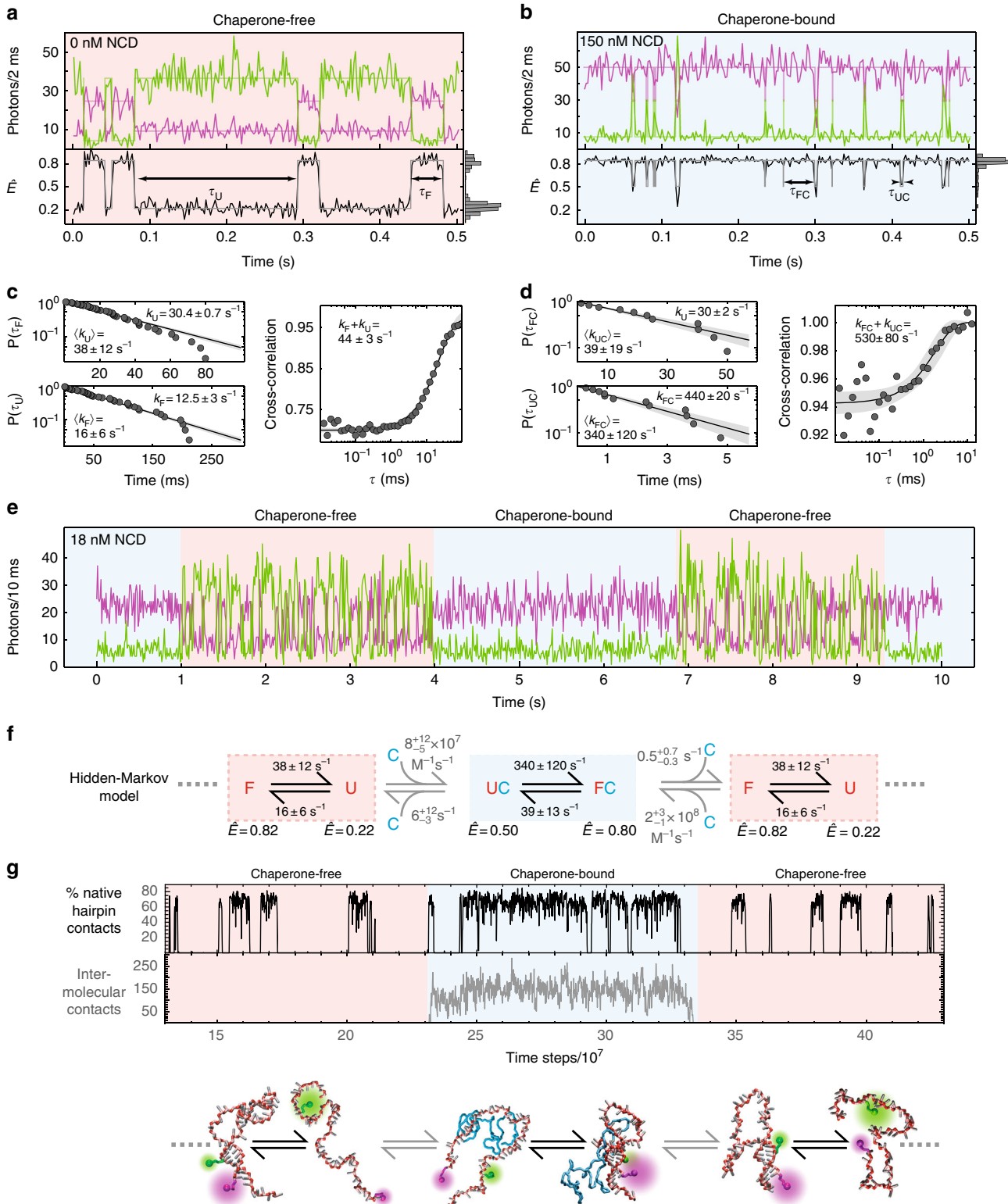

distance dynamics characteristic of intrinsically disordered proteins[32–35], supporting the notion that NCD remains disordered while bound to nucleic acids[19].

To further test the presence of disorder and map out the structural properties of the two binding partners, we placed FRET dyes at different positions within the chaperone and the hairpin (Fig. 3a, c). To isolate the UC complex, analogous experiments were conducted with the folding-incompetent DNA. The intramolecular transfer efficiencies of NCD (S2C-S130C, S2C-

T65C, and T65C-S130C) increase when the UC complex is formed; the same is true for the intramolecular transfer efficiencies of the folding-incompetent DNA (U) (5′-3′, 3′-MID, and 5′-MID) relative to its chaperone-bound counterpart (UC). These results indicate that both the chaperone and the unfolded nucleic acid become more compact when the complex is formed (Fig. 3c), as expected from the mutual screening of the repulsive charges within each chain[36]. A total of 39 intermolecular (Supplementary Fig. 4b, c) and intramolecular (Supplementary

**Fig. 2** Chaperone binding accelerates hairpin folding. **a** Representative donor (green) and acceptor (magenta) fluorescence time traces depicting unassisted folding of the surface-immobilized 5′-3′ FRET-labeled hairpin. The uncorrected transfer efficiency, $\hat{E}$, (black) and the most likely state trajectory (gray) based on the Viterbi algorithm are shown below. **b** Analogous to **a**, but depicting chaperone-assisted hairpin folding with saturating concentrations of NCD (150 nM) to ensure that hairpin molecules are almost always chaperone-associated. **c** Survival probability plots of folded- (upper left) and unfolded-state (lower left) dwell times in the absence of chaperone, fit with single-exponential decays to determine the unfolding and folding rate constants, $k_U$ and $k_F$, respectively; $\langle k_U \rangle$ and $\langle k_F \rangle$ represent mean values obtained from 54 single-molecule measurements. The donor-acceptor fluorescence cross-correlation function (right) decorrelates with a rate constant that is consistent with the sum of $k_U$ and $k_F$ from the dwell-time analysis. **d** Analogous to **c**, but for chaperone-assisted folding; $\langle k_U \rangle$ and $\langle k_F \rangle$ represent mean values obtained from 26 single-molecule measurements. The black lines in **c** and **d** are fits with single exponentials with 99% confidence bands shown in gray. **e** Representative fluorescence time trace measured at a chaperone concentration near $^{app}K_d$. Transitions between chaperone-bound (cyan shading) and chaperone-free (red shading) kinetic regimes arise from binding and dissociation of NCD. **f** Kinetic 4-state model for chaperone-assisted folding. The folded, F, and unfolded, U, conformations of the hairpin freely interconvert in the absence of chaperone, C, with an equilibrium constant that favors U. When the chaperone is bound, the unfolded, UC, and folded, FC, conformations of the hairpin still interconvert, but with an equilibrium constant that favors F. **g** Trajectory from coarse-grained molecular dynamics simulation showing binding (cyan shading) and dissociation (red shading) of the chaperone to the hairpin concomitant with the formation of intermolecular contacts (gray). The folding and unfolding of the hairpin, monitored via native contacts (black), is shifted to favor hairpin formation when the chaperone is bound. Structural representations are taken from the simulations (see Supplementary Movies 1 and 2). Source data for mean values in **c** and **d** are provided as a Source Data File

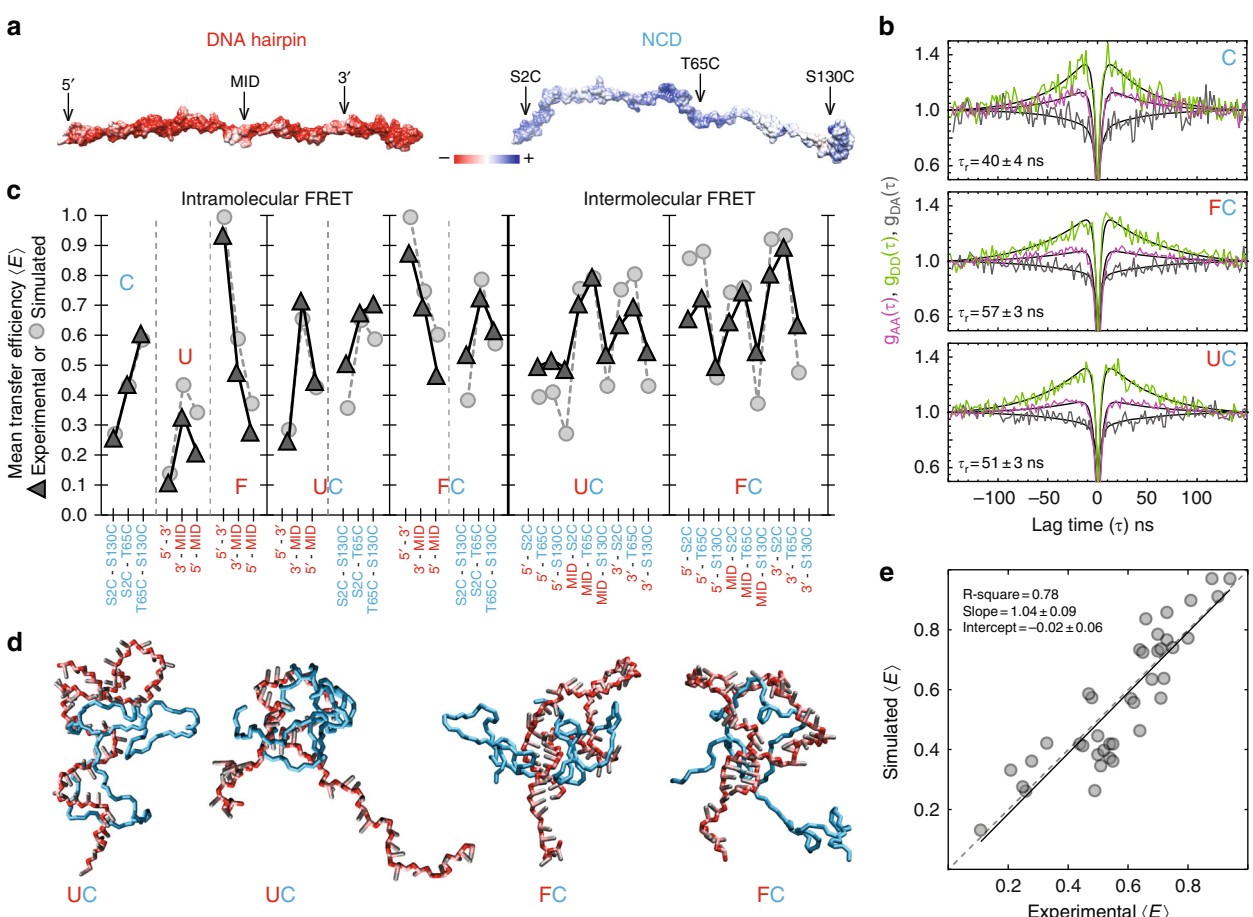

**Fig. 3** Structural properties of chaperone-hairpin complex. **a** Molecular surface representations of the binding partners colored by electrostatic potential (red, negative; blue, positive). Arrows indicate the approximate location of labeling sites. **b** Donor-donor (light green), acceptor-acceptor (magenta), and donor-acceptor (gray) fluorescence correlation functions for FRET-labeled NCD S2C-T65C in isolation (top), bound to the hairpin (middle), and bound to the folding-incompetent hairpin (bottom). Solid black lines are global fits of the three respective correlation functions, which are used to determine the reconfiguration times, $\tau_r$, of the chaperone. **c** Mean transfer efficiencies for all labeling pairs from the mapping experiment (triangles) at 30 mM PB pH 7.0 and the coarse-grained molecular simulations (circles). Mean values are provided in the Source Data file. Labeling positions in the chaperone and the hairpin are shown in cyan and red, respectively. The uncertainties associated with transfer efficiency values are ~0.01 based on multiple simulations and ~0.03 for the experiments (error dominated by instrument calibration). **d** Molecular representations of the disordered nucleic acid-chaperone complex from the simulations (see Supplementary Movies 1 and 2); color code as in Fig. 1a. **e** Correlation between experimental and simulated mean transfer efficiencies. The dashed gray line is the identity line, the solid black line is a linear regression

Fig. 4d) transfer efficiencies were determined using different combinations of donor-acceptor labeling positions (Fig. 3c).

To obtain a molecular description of the two binding partners, we developed a physically-motivated coarse-grained molecular model (see Methods section) of the chaperone and the hairpin (Fig. 3d). Each amino acid is represented by a single bead of appropriate size, and each nucleotide by three beads corresponding to the sugar, base and phosphate groups. The important role of electrostatics for binding is accounted for by Coulomb interactions between all charged groups, including a charge screening term corresponding to the ionic strength of the experimental solution conditions. Other nonspecific attractive interactions and excluded-volume repulsions are described using a short-range potential. Additionally, we included complementary base-pairing interactions within the hairpin allowing for the formation of stem-loop structures (Fig. 1a). Simulations of these coarse-grained models (Supplementary Fig. 7, Supplementary Movies 1 and 2) were used to obtain simulated transfer efficiencies (Supplementary Fig. 8), which we then compared with the corresponding experimental values. By adjusting the three contact energies associated with the model along with a single base-stacking term (see Methods section), simulations of the hairpin-chaperone complex were able to reproduce the 39 measured transfer efficiencies (Fig. 3c) with a root-mean-square deviation of 0.09 (Fig. 3e). The agreement between experimental and simulated data indicates that the behavior of this nucleo-protein complex can be approximated by electrostatically driven interactions between two oppositely-charged and unstructured biopolymers without sequence- or structure-specific interactions (apart from those responsible for the formation of the 7-bp stem). This disordered complex with its electrostatically driven interaction is similar in nature to the recently described high-affinity complex between two oppositely charged disordered proteins[36].

**RNA chaperones as macromolecular counter ions**. In addition to providing information about the structural properties of this nucleoprotein complex, our coarse-grained model also allows us to address the question of how positively charged disordered proteins can function as nucleic acid chaperones. Remarkably, much of the experimentally observed kinetic behavior is reproduced by the simulations (Supplementary Fig. 7), for example: (i) chaperone binding increases the folding rate; (ii) the chaperone binds to the folded and unfolded hairpin with similar rates; and (iii) the chaperone dissociates much more slowly from the folded than from the unfolded hairpin. Together, this behavior results in molecular dynamics trajectories interconverting between two distinct folding regimes (Fig. 2g and Supplementary Fig. 7a) in a way that resembles the experimental data (Fig. 2f). This agreement indicates that our simple molecular model not only accurately describes the conformational properties of a disordered nucleic-acid chaperone that remains unstructured in the bound state, but that it also captures many of the mechanistic properties of chaperone-assisted folding.

Furthermore, the simulations allow us to directly test the hypothesis that NCD-induced compaction of the unfolded nucleic acid causes an increase in the folding rate constant. We thus performed coarse-grained simulations of the isolated hairpin in an external potential chosen to bias its radius of gyration to that of the chaperone-bound hairpin (Supplementary Fig. 9). Indeed, this conformationally biased hairpin also displayed a shift in the equilibrium constant towards the folded state, which primarily resulted from an acceleration of the folding rate constant (Supplementary Fig. 9). Altogether, our results thus suggest an elegant physical mechanism for how a positively charged, intrinsically disordered protein can chaperone conformational transitions in nucleic acids: The chaperone acts as a flexible macromolecular counterion that screens the repulsive negative charges along the phosphate backbone and allows the nucleic acid to more frequently adopt compact (including folded) conformations.

If nonspecific charge screening is the main mechanism involved, then increasing the monovalent salt concentration should have the same qualitative effect on nucleic acid folding as chaperone binding. To test this hypothesis, we performed an analogous set of experiments where NaCl was added to the solution instead of NCD (Supplementary Fig. 10). Indeed, both NCD binding and high concentrations of monovalent salt (i) compact the unfolded hairpin (Fig. 4c), (ii) increase the folding rate constant (Fig. 4d), and (iii) shift the equilibrium to favor folded conformations (Fig. 4a, b). However, the concentrations of monovalent salt required to achieve comparable effects are in the molar range, i.e., seven orders of magnitude greater than for the multivalent chaperone (Fig. 4c, d). This result highlights the remarkable ability of a single disordered chaperone molecule to provide a high local abundance of counterions around the nucleic acid by forming what is essentially a dense macromolecular counterion cloud. Although NCD's macromolecular nature leads to much higher affinity and quantitatively stronger effects, its chaperoning function is qualitatively consistent with the trend observed for the facilitated folding of nucleic acids using a variety counterions with increasing charge and size, ranging from metal ions to polyamines[37–39].

## Discussion

The RNA and DNA hairpins used in this report represent a structural motif that is ubiquitous in folded nucleic acids, and therefore the functional mechanism we propose is expected to be applicable to a wide range of chaperone-assisted conformational transitions in nucleic acids. The extent of its applicability certainly merits further investigation. For instance, it has been proposed that hepatitis C virus genome dimerization is initiated by forming intermolecular base pairs between the unpaired nucleotides in the stem-loop structure of the dimer linkage sequence (DLS; Supplementary Fig. 1), giving rise to what is commonly referred to as RNA kissing loops[40]. It is plausible that NCD promotes kissing loop formation, and therefore genome dimerization, via the charge screening mechanism we describe for chaperone-assisted hairpin formation: NCD will bind to the DLS, serving as a macromolecular counter ion and reducing the effective charge of the nucleoprotein complex, which reduces the repulsion between the two interaction sites (i.e., the unpaired nucleotides in the loops of the DLS). As a result, the rate constant for kissing loop formation would be expected to increase, and the equilibrium would shift to favor genome dimerization after binding of NCD.

Intrinsically disordered RNA chaperones that bind nucleic acids with high affinity thus represent an efficient and genetically controllable strategy for altering the conformations and dynamics of functional nucleic acids (both DNA and RNA) in a cellular environment, whereas factors such as cellular salt concentrations can only be regulated within narrow bounds. These types of interactions are a likely reason why nucleic acids exposed to cellular extracts are bound to a variety of proteins[41] and why the structural and dynamic properties of RNA in cellular environments tend to differ from those in vitro[42,43]. Furthermore, the lack of sequence and structural specificity of the macromolecular counterion mechanism described here suggests that other positively charged intrinsically disordered viral proteins, such as NCp7[44] and Tat[44,45] from HIV-1, as well as the core protein from the West Nile virus[46], might employ a similar mode of action. In

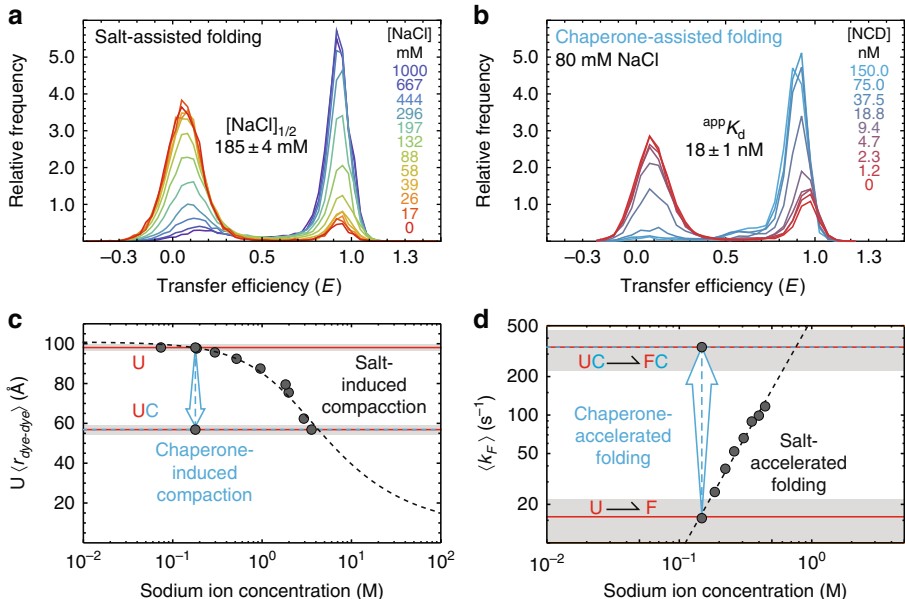

**Fig. 4** NCD functions as a macromolecular counterion. Transfer efficiency histograms for **a** salt-assisted and **b** chaperone-assisted folding of the 5′-3′ FRET-labeled DNA hairpin. Increasing concentrations of either NaCl or NCD promote formation of the hairpin, but the transition midpoint concentrations (i.e., $^{app}K_d$ and $[NaCl]_{1/2}$) for the two species differ by a factor of ~$10^7$. **c** Mean inter-dye distance for the folding-incompetent hairpin as a function of sodium ion concentration. Source data for mean values are provided as a Source Data file. The compaction of the unfolded nucleic acid induced by chaperone binding (cyan arrow) is comparable to that present at molar concentrations of NaCl. The dashed black line results from a fit to a worm-like chain with a salt-dependent (i.e., electrostatic) persistence length (see Methods section). **d** Mean folding rate constant for hairpin formation as a function of NaCl concentration. Source data for mean values are provided as a Source Data file. Chaperone binding gives rise to a folding rate constant comparable to solutions containing molar concentrations of NaCl (cyan arrow). The extrapolation (dashed black line) is a fit of the data to a power law of the form $f(x) = ax^n$, where $x$ is the concentration of NaCl, and $n$ is related to the number of counterions taken up upon forming the transition state[62]

addition to modulating the affinity of nucleic-acid-binding proteins[47], positively charged intrinsically disordered regions may also represent a way to append this type of chaperone activity to transcription factors[48], nucleocapsid proteins[15], and histone-like proteins[49].

Although macromolecular charge screening is likely to be a widely applicable mechanism of intrinsically disordered RNA chaperones, there is of course a range of complementary mechanisms. For example, the structurally more well-defined Moloney murine leukemia virus nucleocapsid protein preferentially interacts with guanosine residues, preventing them from forming non-native contacts that would otherwise impede formation of the native state[14]. The hexameric bacterial RNA chaperone Hfq makes use of multiple intrinsically disordered tails to facilitate release of RNA target molecules[50], while other RNA chaperones with ATP-dependent helicase activity unwind their substrates, leading to an iterative annealing mechanism[9]. Despite the wide range of mechanisms employed by RNA chaperones, many of them are likely to complement the macromolecular counterion mechanism we propose here and may therefore cooperate in facilitating conformational transitions for a wide range of DNA and RNA molecules.

## Methods

**Preparation and labeling of nucleic acids**. Both the hairpin (HP) and the non-complementary, folding-incompetent hairpin (ncHP) were purchased as chemically synthesized deoxyoligonucleotides (Integrated DNA Technologies). One or two of the following chemical moieties were introduced during synthesis to facilitate dye attachment: a 5′ dithiol at position 0 (i.e., the phosphate preceding the first cytosine) for maleimide labeling, an amino-modified C6 dT in the middle of the loop at position 27, and an amino-modified C6 dT towards the 3′ end (position 48) of the hairpin for succinimidyl ester labeling. In total, four different oligonucleotides were used to generate the seven different labeling permutations for HP; the same is true for ncHP (Supplementary Table 1). The RNA-based HP had a

sequence similar to the DNA HP with similar synthetic modifications (Supplementary Table 2).

Prior to labeling, the synthetically incorporated thiol groups were reduced using 5 mM tris(2-carboxyethyl)phosphine. Either N-hydroxysuccinimide-functionalized or maleimide-functionalized donor (Cy3B; GE Healthcare) and acceptor (CF660R; Biotium) dyes were site-specifically incorporated at the desired positions in the oligonucleotides, which contained either primary amines or thiols, using the manufacturers' suggested protocols. The labeled constructs were purified on a reversed-phase C18 column (Dr. Maisch) using a high-performance liquid chromatography (HPLC) system (Agilent 1100 series). Samples were lyophilized overnight, then dissolved in 50 mM sodium phosphate buffer (PB) pH 7.0 and stored at −80 °C until use.

**Preparation and labeling of NCD**. Gene variants of a clinical isolate of NCD (GenBank: CAE46584.1) were cloned into a pET-47b(+) vector, transfected into BL21 (DE3) *E.coli* cells (New England Biolabs). After 3 h of growth at 37 °C, recombinant expression was induced via 0.5 mM isopropyl β-D-1-thiogalactopyranoside for a total of for 3 h at 37 °C[19]. In the sequence, the terminal (134) tryptophan residue was removed to prevent dye quenching; the only natural cysteine at position 98 was changed to methionine; and either one or two cysteine residues were introduced via site-directed mutagenesis at the desired labeling positions (S2, T65, and S130), resulting in 6 different amino acid sequences (Supplementary Table 3). The correct mass of all proteins was confirmed via electrospray ionization mass spectrometry (Functional Genomics Center Zurich). See Supplementary Table 7 for the sequences of all primers used in this work.

Immobilized metal-ion affinity chromatography and HPLC were used to purify the recombinantly expressed proteins. The proteins were then stochastically labeled with maleimide-functionalized donor and/or acceptor dyes and purified on a reversed phase C18 column (Dr. Maisch) using HPLC to remove double donor and double acceptor species. Samples were lyophilized overnight, then dissolved in 50 mM sodium phosphate buffer (PB) pH 7.0 and stored at −80 °C until use.

**Free-diffusion single-molecule FRET spectroscopy**. Free-diffusion experiments were performed on a modified version of a confocal single-molecule instrument (MicroTime 200, PicoQuant) using pulsed interleaved excitation[51,52] and time-correlated single photon counting. Direct excitation of the donor was accomplished using 100 μW (measured at the back aperture of the microscope objective) of filtered light (520 ± 5 nm) from a pulsed (20 MHz) continuum laser (SC400, Fianium), which also triggered the 635-nm pulsed diode laser (PicoQuant) that directly excited (100 μW) the acceptor. Prior to being detected on one of four

avalanche photodiodes (AQRH-series, PerkinElmer), photons were separated by polarization and color using a polarizing beamsplitter cube (PicoQuant) and two dichroic mirrors (Chroma), respectively.

All photon arrival time records were binned at 0.5 ms. The background-corrected numbers of acceptor ($\hat{N}_A^d$) and donor ($\hat{N}_D^d$) photons resulting from direct donor excitation were used to calculate the uncorrected transfer efficiency, $\hat{E} = \frac{\hat{N}_A^d}{\hat{N}_A^d + \hat{N}_D^d}$, for each burst, and the total numbers of photons resulting from donor ($\hat{N}_{tot}^d$) and acceptor ($\hat{N}_{tot}^a$) direct excitation were used to determine the apparent fluorescence stoichiometry ratio, $\hat{S} = \frac{\hat{N}_{tot}^d}{\hat{N}_{tot}^d + \hat{N}_{tot}^a}$[53–55]. Only bursts resulting from molecules with active donor and acceptor dyes ($0.3 < \hat{S} < 0.7$) were used to construct $\hat{E}$ histograms. Correction factors for cross talk, direct excitation, as well as differences in excitation and detection efficiencies were calculated from the mean $\hat{E}$ and $\hat{S}$ values of the three populations present in the sample (active FRET pair at $\hat{S} \approx 0.5$, active donor only at $\hat{S} \approx 1.0$, and active acceptor only at $\hat{S} \approx 0.0$) and used to calculate corrected values for $E$, the transfer efficiency, and $S$, the stoichiometry ratio[53–55]. All data analysis was conducted using a custom Wolfram Symbolic Transfer Protocol (WSTP) add-on for Mathematica (Wolfram).

All free-diffusion experiments were conducted with labeled biomolecule concentrations ranging between 50 and 250 pM in one of two buffers systems: either standard buffer (50 mM PB pH 7.0 and 0.01% Tween20) with the specified concentration of NaCl, or low-salt buffer (30 mM PB pH 7.0 and 0.01% Tween20). Ascorbic acid and methylviologen were included at a final concentration of 1 mM each to improve the performance of the donor and acceptor dyes[56]. For intermolecular FRET experiments conducted in low-salt buffer, we occasionally observed 1:2 hairpin-chaperone complexes (Supplementary Fig. 4b). In these cases, we only consider the transfer efficiency of the 1:1 species (at $S \approx 0.5$) because we were primarily interested in the conformational properties of the 1:1 complex and because interpreting transfer efficiencies in the presence of multiple acceptors can be difficult[57]. For intramolecular FRET experiments on the FRET-labeled hairpin conducted in low-salt buffer, we work at sub-saturating concentrations of chaperone to minimize the interference of these 1:2 complexes with our transfer efficiency measurements.

**Recurrence analysis**. For recurrence analysis[29], experiments were conducted on a custom-built confocal single-molecule fluorescence instrument, where 100 µW of light from a continuous-wave 532-nm laser (LaserBoxx 532, Oxxius) was used to excite the donor, and 25 µW of light from a 5-MHz 635 nm pulsed laser diode (PicoQuant) was used to excite the acceptor. Photons were detected using four single-photon avalanche diodes (AQRH-series, PerkinElmer) and recorded using a time-correlated single-photon counting module (HydraHarp, PicoQuant). For the analysis, photon arrival times were binned in 250-µs bins. Only bins containing fluorescence from both donor and acceptor dyes were considered as bursts for recurrence analysis. Additional details regarding recurrence analysis can be found elsewhere[29]. All data analysis was conducted using a custom WSTP add-on for Mathematica (Wolfram).

**Surface-immobilization experiments**. All surface-immobilization experiments of FRET-labeled HP were conducted on the MicroTime 200 (PicoQuant) used for the free diffusion measurements, but 1 µW (measured at the back aperture of the microscope objective) of light from a continuous-wave 532-nm laser (LaserBoxx 532, Oxxius) was used to excite the donor since pulsed excitation led to lower photostability. Biotin-neutravidin binding was used to non-covalently immobilize the biotin-modified nucleic acids to biotinylated, polyethylene glycol-passivated quartz cover slides (MicroSurfaces) at a surface density of less than one molecule per square micrometer. All surface-immobilized experiments were performed in solutions containing 50 mM phosphate buffer (PB) pH 7.0 with 80 mM NaCl. The presence of a single-step photobleaching event was used to confirm the presence of single molecules. The resulting fluorescence time traces were individually inspected to identify segments without fluorophore blinking. A maximum-likelihood (MLH) analysis of the donor and acceptor photon arrival times from these segments[58] was used to identify the most likely rate constants for a hidden Markov model of our data. The likelihood ($L$) function can be written as:

$$L = \mathbf{1}^T \prod_{k=2}^{N_{ph}} \left( \mathbf{F}(c_k) e^{(\mathbf{K}-\mathbf{N})\tau_k} \right) \mathbf{F}(c_1) \mathbf{p}_{eq} / \mathbf{1}^T \mathbf{N} \mathbf{p}_{eq}, \qquad (1)$$

where $\mathbf{1}^T = (1,1,1,1)$ is the transposed unity vector, $N_{ph}$ is the total number of photons in the trajectory, and $\mathbf{F}(c_k)$ is a matrix that depends on the "color" $c_k$ of the $k$th photon. For $\mathbf{F}(acceptor) = \mathbf{N}_A$ and for $\mathbf{F}(donor) = \mathbf{N}_D$. Here, $\mathbf{N}_A$ and $\mathbf{N}_D$ are diagonal matrices with the acceptor, $n_{A_i}$, and donor, $n_{D_i}$, count rates associated with state $i$ on the diagonal. The matrix $\mathbf{N} = \mathbf{N}_A + \mathbf{N}_D$, with diagonal elements $n_i = n_{A_i} + n_{D_i}$. $\mathbf{K}$ describes the transition rates between states $i$ and $j$. The inter-photon time of the $k$th photon is represented by $\tau_k = t_k - t_{k-1}$, where $t_k$ is the detection time of the $k$th photon. The vector $\mathbf{p}_{eq}$ contains the equilibrium populations of the various states in the hidden Markov model (i.e., $\mathbf{K}\,\mathbf{p}_{eq} = 0$). For the analysis of trajectories in the presence and absence of 150 nM NCD, the likelihood function of each fluorescence time trace was maximized by adjusting the parameters contained

within $\mathbf{N}_A$, $\mathbf{N}_D$, and $\mathbf{K}$ for a two-state hidden-Markov model (i.e., either U ⇌ F or UC ⇌ FC).

To visually assess the quality of the MLH analysis, the Viterbi algorithm was used in conjunction with the values of $\mathbf{N}_A$, $\mathbf{N}_D$, and $\mathbf{N}$ from the MLH analysis to determine the most likely state trajectory, which allowed us to determine dwell times. Finally, survival probability distributions determined from the dwell times were fit to exponential decay functions to obtain rate constants (e.g., $k_U$ and $k_F$), which were indistinguishable from MLH-derived values within experimental error. Additionally, the donor-acceptor fluorescence cross-correlation signal between $\tau = 10^{-2}$ and $\tau = 10^2$ ms was fit to a single-exponential function to determine the relaxation rate constants, $k = k_U + k_F$, which were also consistent with the dwell-time and MLH analyses.

For experiments conducted at concentrations of unlabeled NCD near the apparent dissociation constant (i.e., 18 nM), 0.33 µW of light was used to excite the donor to reduce photobleaching, thereby enabling longer observation times of surface-immobilized FRET-labeled molecules. Additionally, a dwell-time analysis like the one described above was not possible because: (i) the transfer efficiency of the F and FC states were very similar and (ii) both these states were populated under these conditions. Therefore, we used the MLH routine to analyze all fluorescence time traces with a four-state hidden Markov model to determine the binding and dissociation rate constants for the interaction of NCD with the folded and unfolded conformations of the hairpin (Fig. 2f, gray). All other parameters in the four-state hidden Markov model (i.e., $\mathbf{N}_A$ and $\mathbf{N}_D$ as well as the hairpin folding and unfolding rate constants, $k_U$, $k_F$, $k_{UC}$, and $k_{FC}$) were fixed (Fig. 2f, black) to values obtained in experiments conducted on immobilized hairpin molecules in the presence and absence of 150 nM unlabeled NCD (Fig. 2a, b). All data analysis was conducted using a custom WSTP add-on for Mathematica (Wolfram).

**Nanosecond fluorescence correlation spectroscopy**. To determine the reconfiguration dynamics of NCD, we used a variant stochastically labeled with Alexa 488 (donor) and Alexa 549 (acceptor) at positions 2 and 65. Prior to measurements, the FRET-labeled protein was purified using HPLC to remove the undesired double-donor and double-acceptor populations. Correlation functions were generated from ~10 h measurements using 100 µW of 488-nm continuous-wave excitation (measured at the back aperture of the microscope objective) to directly excite the donor. All measurements contained approximately 50 pM FRET-labeled NCD in low-salt buffer. Only photons arising from the FRET-labeled population (i.e., $\hat{E} > 0.2$) were considered for the correlation analysis. Lag times ranging between $\tau = -150$ ns and $\tau = 150$ ns from the acceptor autocorrelation function, $g_{AA}(\tau)$, the donor autocorrelation function, $g_{DD}(\tau)$, and the donor-acceptor crosscorrelation function, $g_{DA}(\tau)$, were globally fit to a model with two exponential components for each correlation:

$$g_{ij}(\tau) = a_{ij} \left( 1 + A_{AB_{ij}} e^{-\left| \frac{\tau - t_{0_{ij}}}{\tau_{AB_{ij}}} \right|} \right) \left( 1 + A_{CD_{ij}} e^{-\left| \frac{\tau - t_{0_{ij}}}{\tau_{CD}} \right|} \right), \qquad (2)$$

where the amplitude $a_{ij}$ depends on the mean number of molecules in the confocal volume, triplet blinking, and the background signal. Small differences in the temporal alignment of the donor and acceptor channels are accounted for with the time offset, $t_{0_{ij}}$. In this model, $\tau_{AB_{ij}}$ is a fast (<5 ns), anticorrelated component (i.e., $A_{AB_{ij}} < 0$) for photon antibunching (AB), and $\tau_{CD}$ is a slower (>10 ns) component shared by all correlations and corresponding to chain dynamics (CD). This slower component is anti-correlated in the donor-acceptor (DA) crosscorrelation (i.e., $A_{CD_{DA}} < 0$), as expected for fluctuations in transfer efficiency due to distance dynamics. The reconfiguration time ($\tau_r$) we report accounts for the fact that $\tau_{CD}$ does not directly represent the decay of the end-to-end distance correlation function; additional details regarding the nanosecond correlation spectroscopy are presented elsewhere[33,59,60]. All data analysis was conducted using a custom WSTP add-on for Mathematica (Wolfram).

**Analysis of binding equilibria**. All binding equilibria were analyzed using a 2-state Hill-type binding isotherm[61] with the following functional form:

$$f(x) = f_i + \frac{(f_f - f_i)\, x^n}{x^n + \varphi^n}, \qquad (3)$$

where $f$ is the experimental observable (e.g., $\langle E \rangle$ or fraction bound), and $x$ the experimental variable (e.g., NaCl or NCD concentration). The fit parameters $f_i$ and $f_f$ correspond to the asymptotic limits of $f(x)$ at low and high concentration, respectively. The fit parameter $\varphi$ represents the midpoint (e.g., apparent dissociation constant) of the transition from $f_i$ to $f_f$ and is defined as $\frac{f(\varphi) - f_i}{(f_f - f_i)} = \frac{1}{2}$. Finally, the fit parameter $n$ corresponds to the sensitivity of the transition to concentration and is defined as $n = (\partial \ln K / \partial \ln x)$, where $K$ corresponds to the apparent equilibrium constant. When the experimental variable is salt (e.g., NaCl), as in the case of Supplementary Fig. 3, this formalism can be used to interpret $n$ in terms of either counterion release ($n < 0$) or counterion uptake ($n > 0$)[62]. All data analysis was conducted using Mathematica (Wolfram).

**Coarse-grained representation and energy function**. In the coarse-grained model, each residue of the protein NCD was represented by a single bead (of type C) located at the position of the alpha-carbon atom, while the DNA was represented by three beads per residue, one for the ribose (R bead), located at the C4′ position, one for the phosphate at the P position (P bead), and one for the base. The base bead (B) was located at the average position of the N1 of adenine, the N1 of guanine, the N3 of thymidine and the N3 of cytidine in an idealized DNA duplex (see below for further geometrical details). The energy function was given by a standard force field expression:

$$
\begin{aligned}
E = &\sum_{(i,j)\in\text{bonds}} \tfrac{1}{2}k_{\text{b}}\left(r_{ij}-r_{ij,0}\right)^2 + \sum_{(i,j,k)\in\text{angles}} \tfrac{1}{2}k_{\theta}\left(\theta_{ijk}-\theta_{ijk,0}\right)^2 \\
&+ \sum_{(i,j)\in\text{stack}} 4\varepsilon_{\text{stack}}\left(\left(\frac{\sigma_{\text{stack}}}{r_{ij}}\right)^{12}-\left(\frac{\sigma_{\text{stack}}}{r_{ij}}\right)^{6}\right) \\
&+ \sum_{(i,j)\notin\text{native}} 4\varepsilon_{ij}\left(\left(\frac{\sigma_{ij}}{r_{ij}}\right)^{12}-\left(\frac{\sigma_{ij}}{r_{ij}}\right)^{6}\right) \\
&+ \sum_{(i,j)\in\text{native}} 4\hat{\varepsilon}_{ij}\left(\left(\frac{\hat{\sigma}_{ij}}{r_{ij}}\right)^{12}-\left(\frac{\hat{\sigma}_{ij}}{r_{ij}}\right)^{6}\right) + \sum_{i,j}\frac{q_iq_j}{4\pi\epsilon_0 D r_{ij}}\exp\left[-\frac{r_{ij}}{\lambda_{\text{DH}}}\right]
\end{aligned}
\tag{4}
$$

The first two terms of Eq. 4 account for the covalently bonded structure of the chain, with $r_{ij}$ being the distance between atoms $i$ and $j$, $k_{\text{b}}$ the harmonic bond force constant, $\theta_{ijk}$ the angle between bonded atoms $i$, $j$, and $k$, and $k_{\theta}$ is the harmonic angle force constant. The ideal values of the bond lengths and angles ($r_{ij,0}$ and $\theta_{ijk,0}$ respectively) and the force constants are given in Supplementary Table 4. The third term is a stacking energy that is defined for all base beads (B) that are adjacent in sequence, irrespective of whether they are involved in forming duplex structure. The fourth and fifth terms of Eq. 4 correspond, respectively, to a generic non-bonded term for atom pairs not in contact in the native DNA duplex and not involved in stacking, and a Gō-like (structure-based) term for those atom pairs in contact in the DNA duplex. The generic non-bonded term is a function of the parameters $\varepsilon_{ij}$ and $\sigma_{ij}$, which depend only on the types of beads interacting (i.e., protein bead C, or P, R or B bead of DNA). The corresponding interaction matrix is given in Supplementary Table 5. The Gō-like term is defined for beads which form base pairs in the duplex structure, and was only used for the DNA sequence capable of forming a duplex. The set of native interaction parameters $\hat{\varepsilon}_{ij}$ and $\hat{\sigma}_{ij}$ for each pair of base-paired residues is given in Supplementary Table 6. Note that, by symmetry, analogous native contacts are implicitly defined with $j$ and $i$ permuted. The native contact distances are based on idealized B-form Watson–Crick DNA. The contact energies listed above are those used for simulating equilibrium folding dynamics of the DNA; for simulating the folded state, a value of 5.0 kJ mol$^{-1}$ was used to keep the DNA folded. The last term in Eq. 4 is the coulombic potential: Every bead representing a charged amino acid in the chaperone is assigned the appropriate net charge ($q_i$), and the DNA phosphate beads are each given a charge of $-1$. Electrostatic forces are screened with a Debye–Hückel potential with Debye length $\lambda_{\text{DH}}$ determined from the relevant ionic strength. The parameters $\epsilon_0$ and $D$ are, respectively, the permittivity of free space and the relative dielectric of the medium, taken to be 80. Note that the only sequence-specific features of the protein in the model are the residue charges. For the DNA, the only sequence-dependent feature is the presence of the complementary strands in the foldable sequence, which are included via the structure-based term.

**Comparing simulations and experiments**. Transfer efficiencies were calculated from the coarse-grained molecular dynamics simulations using the distribution of inter-residue distances. Because the reconfiguration dynamics of the complex occur on a 50-ns timescale, we assume that the distribution of inter-residue distances, $P(r)$, is sampled on a time scale that is much slower than the fluorescence lifetime of the donor (e.g., 3 ns) and that the rotational correlation time of the dyes is much faster than the fluorescence lifetime of the donor. This gives rise to the following expression for the mean transfer efficiency:[63] $E = \int_0^\infty E(r)P(r)dr$, where $E(r) = \left(1+(r/R_0)^6\right)^{-1}$ and $R_0 = 59$ Å is the Förster radius of the Cy3B-CF660R FRET pair.

**Model parameter optimization**. The bonded parameters (bonds, angles) were based, in the case of the protein, on an extended random-coil structure, and in the case of the DNA, on an idealized Watson–Crick duplex structure. These parameters were fixed and not subject to optimization. The non-bonded radius for all protein and DNA beads was also fixed to 0.6 nm. This value is approximately consistent with the volume of a typical amino acid, ribose or base. The single protein-protein non-bonded energy parameter was adjusted to match the FRET efficiencies of the protein in isolation and was fixed to that value in all subsequent simulations. The obtained value of 0.15 $k_{\text{B}}T$ is almost identical to the 0.16 $k_{\text{B}}T$ used for protein-protein interactions in a closely related model we used to interpret recent FRET experiments on the interaction of two highly charged proteins[36]. For the DNA, the non-bonded interaction energy $\varepsilon_{ij}$ was set to a small value for most beads, except for adjacent bases, which are known to form favorable base stacking interactions. The single stacking pair energy parameter between sequence-neighboring bases was optimized in order to reproduce the FRET efficiencies of the isolated, folding-incompetent DNA

(without native interactions defined). This led to an average stacking free energy of ~8.79 kJ mol$^{-1}$, which is consistent with other estimates of the free energy gain by base stacking[64]. This parameter was then fixed for all simulations involving DNA. The last parameter optimized to match FRET data was a single protein-DNA interaction energy, common to all DNA beads, which was determined to match the intermolecular FRET data for the complex of NCD with the unfolded DNA. No adjustable parameters were added for the folded DNA: for comparison with experimental FRET efficiencies of the folded DNA in isolation, or in complex with NCD, the strength of each duplex native contact was set to a sufficiently large value (5.0 kJ mol$^{-1}$) that the DNA was essentially always folded. For simulations of equilibrium folding dynamics, a smaller native contact energy of 3.279 kJ mol$^{-1}$ was used.

**Simulation details**. For most simulations, unbiased Langevin dynamics were run at 300 K with Gromacs[65] version 5.1.4 or 4.0.2 (www.gromacs.org), using a time step of 10 fs and a friction coefficient of 0.1 ps$^{-1}$. All bond lengths were constrained to their initial values. A cubic box of side 40 nm was used for all simulations of equilibrium folding dynamics of the DNA. Folding kinetics were estimated from long equilibrium simulations of the NCD and DNA together. A four-state kinetic model comprising the states U, F, UC, and FC (Fig. 2f) was fitted to the simulation data by assigning transitions using a transition-based or core state assignment[66]. The trajectory was initially unassigned until it reached one of the core states. It then only changed assignment once it reached a different core state. The core sets were defined as Core$_{\text{UC}}$ = Unfolded $\cap$ Bound, Core$_{\text{FC}}$ = Folded $\cap$ Bound, Core$_F$ = Folded $\cap$ Unbound, Core$_U$ = Unfolded $\cap$ Unbound. The folded and unfolded sets were defined as $\{x : Q(x) > 0.7\}$ and $\{x : Q(x) < 0.02\}$, respectively, with $Q(x)$, the fraction of native hairpin contacts for conformation $x$, defined as:

$$
Q(x) = \sum_{(i,j)\in\text{native}} \frac{1}{1+\exp\left[\beta\left(r_{ij}(x)-\gamma\sigma_{ij}\right)\right]}
\tag{5}
$$

Here, the sum runs over DNA native contact pairs $(i, j)$ defined as in Supplementary Table 6, and the constants $\beta$ (set to 50 nm$^{-1}$) and $\gamma$ (set to 1.4), respectively, control the steepness of the switching function and account for fluctuations within the native state. The Bound core set was defined by $\{x : d_{\min}(x) < 0.7 \text{ nm} \wedge n_{\text{inter}}(x) > 70\}$, where $d_{\min}(x)$ is the minimum of all pair distances between protein and DNA beads in conformation $x$, and $n_{\text{intex}}(x)$ is the number of intermolecular contacts (i.e., number of protein-DNA atom pairs less than 1.0 nm apart). Similarly, the unbound set was defined as $\{x : d_{\min}(x) > 4 \text{ nm} \wedge n_{\text{inter}}(x) < 1\}$. Rate coefficients for transitions between states were determined using a lifetime-based method[66].

To mimic the confining effect of the chaperone on the DNA, a bias potential was obtained by fitting a smooth curve $V_{\text{bias}}$ to the difference between the free energy surface projected onto the radius of gyration, $r_{\text{g}}$ of the isolated DNA and the DNA in complex with the chaperone (note that we follow the convention here of using lowercase $r_{\text{g}}$ to refer to the radius of gyration of an individual conformation, and uppercase $R_{\text{g}}$ for the ensemble average). The exact bias used was (with $V_{\text{bias}}$ in units of kJ/mol and $r_{\text{g}}$ in nm):

$$
V_{\text{bias}}\left(r_{\text{g}}\right) = \begin{cases} -49.302 + 27.525r_{\text{g}} - 4.7448r_{\text{g}}^2 + 0.27657r_{\text{g}}^3 & \text{for } r_{\text{g}} < 5.7186 \\ 2.4204 + 0.39127r_{\text{g}} & \text{otherwise} \end{cases},
\tag{6}
$$

which is continuous and differentiable everywhere. This bias was then applied as an external potential to the radius of gyration of the DNA without chaperone using PLUMED[67,68] (www.plumed.org). The potential was implemented using cubic splines with control points defined every 0.005 nm using the above equation. The folding equilibrium and kinetics of the DNA in the presence and absence of this external potential were compared to the corresponding kinetics in the presence of the NCD. A similar lifetime-based kinetic model was used, in this case with only two states defined using folded and unfolded as core sets.

**Salt-induced compaction of unfolded hairpin**. The salt-induced compaction of the unfolded DNA is fit using a worm-like chain model[69–71], which describes the end-to-end distance ($r$) distribution of a polymer chain in terms of the persistence length ($l_{\text{p}}$), the contour length ($l_{\text{c}}$), and a normalization constant ($C$):

$$
P(r) = \frac{4\pi C r^2}{l_{\text{c}}^2\left[1-(r/l_{\text{c}})^2\right]^{9/2}}\exp\left(-\frac{3l_{\text{c}}}{4l_{\text{p}}\left[1-(r/l_{\text{c}})^2\right]}\right).
\tag{7}
$$

Odijk-Skolnick-Fixman (OSF) theory[72,73] describes how the electrostatic component of the persistence length ($^{\text{elec}}l_{\text{p}}$) of a polyelectrolyte depends on the ionic strength of the solution via $^{\text{elec}}l_{\text{p}} = l_{\text{B}}/4\kappa^2 A^2$, where $l_{\text{B}}$ is the Bjerrum length, $\kappa^{-1}$ is the Debye length, and $A$ is the effective segment length in the absence of electrostatics. The total effective persistence length is thus $l_{\text{p}} = {}^{\text{elec}}l_{\text{p}} + A$, and the contour length of the chain is $l_{\text{c}} = NA$, where $N$ is the number of effective chain segments. Since only the values of $N$ and $A$ are unknown for our unfolded DNA, they are treated as fit parameters, where any given combination of $N$ and $A$ gives rise to a unique salt dependence of $\langle r \rangle$. Although the best fit gives rise to a chain with many ($N = 210 \pm 10$) small segments ($A = 0.05 \pm 0.01$ nm), the product of

these two values is 10.5 nm, which is a reasonable contour length for our single-stranded nucleic acid.

**Reporting summary**. Further information on research design is available in the Nature Research Reporting Summary linked to this article.

## Data availability

Data supporting the findings of this manuscript are available from the corresponding authors upon reasonable request. A reporting summary for this Article is available as a Supplementary Information file. The source data underlying mean values and gel images in Figure panels 1c, d, 2a–d, 3c, e, and 4c, d in addition to Supplementary Figure panels 1c, 2a, b, 4a, 5b, 6a, b, and 10b, c, f are provided as a Source Data file.

## Code availability

Example input files for running the coarse-grained model in Gromacs (www.gromacs. org) are provided in a compressed archive (initial coordinates, topology files, run input and auxiliary files, and a summary README file) as Supplementary Software (Simulation_Models.tgz). A custom WSTP add-on for Mathematica (Wolfram Research) used for the analysis of single-molecule fluorescence data is available upon request.

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

## Acknowledgements

The authors thank Franziska Zosel for helpful discussions regarding data analysis and the Functional Genomics Center Zurich for expert mass spectrometry analysis. Support for this work was provided by the Swiss National Science Foundation (to B.S.) and the European Molecular Biology Organization (to E.D.H., ATLF 471-2015). R.B.B. was supported by the Intramural Research Program of the NIDDK at the National Institutes of Health. This work utilized the computational resources of the NIH HPC Biowulf cluster (http://hpc.nih.gov).

## Author contributions

The project was conceptualized and supervised by E.D.H., R.B.B. and B.S. Data curation and analysis was carried out by E.D.H., Z.L. and R.B.B. Research funds were acquired by E.D.H., R.B.B. and B.S. Scientific investigations were conducted by E.D.H., Z.L. and R.B.B., with methodological developments provided by E.D.H., D.N., R.B.B. and B.S. Administrative aspects of the project were overseen by E.D.H. and B.S. The project resources were provided by R.B.B. and B.S. Software development was carried out by E.D.H., D.N. and R.B.B. Data validation was overseen by E.D.H., R.B.B. and B.S. Visualization of data was performed by E.D.H. and R.B.B. The original draft was written by E.D.H. with edits and revisions provided by E.D.H., D.N., R.B.B. and B.S.

## Additional information

**Competing interests:** The authors declare no competing interests.

