## [Peer Review File · Nature Communications]

Reviewers' Comments:

Reviewer #1:

Remarks to the Author:

Holmstrom et al. present an interesting analysis of how intrinsically disordered proteins can act as nucleic acid folding chaperones. This question is particularly interesting given that IDPs or IDRs are present widely in nucleic acid binding and processing proteins. Furthermore, the HCV system is relevant for biology and disease. The authors carefully carry out a range of single molecule fluorescence experiments and find that the highly charged NCD domain of HCV protein increases the efficiency of folding of nucleic acid stem-loop structures, as well as generic compaction. Furthermore, the NCD itself is compacted yet disordered in the complex, reminiscent of IDP-protein complexes studied by the same lab and others in the field. The authors also carry out coarse-grained MD simulations that recapitulate key results and support the electrostatic model. The work is very well done, using a range of experiments and labeled constructs, and new and interesting information about an important biological problem is revealed from the data and simulations. Thus, the work will be of high interest for the readers of this journal. I only have a couple of minor comments that the authors can address to improve the manuscript.

As the authors mention early on, RNA folding has been shown to often result in trapped/misfolded intermediates. I would think that the stabilization of compact states shown here would often increase the barrier from these trapped states to native structures. Is this correct? If so, can the authors comment on how the systems might get around this problem?

As far as I can see, the results here are on DNA folding. Authors, please comment more on the expected differences and potentially more complex behavior for RNA folding.

Reviewer #2:

Remarks to the Author:

Holmstrom et al demonstrate how an intrinsically disordered protein can function as an RNA chaperon. By forming only unspecific electrostatic interaction the chaperon efficiently screens the RNA charges smoothing the free energy landscape and increasing the folding speed and folded state population.

The mechanism is shown to be equivalent to the addition of 1M of NaCl.

A simple physical model is used to simulate the protein-RNA interaction well capturing the observed experimental observation and further demonstrating that the molecular recognition is unspecific and that the resulting effect on the RNA is that of decreasing its average size.

The work is extremely well done and definitely worth publishing.

The only minor observation I feel to add is that

- GROMACS and PLUMED should be referenced
- It would be nice to have a reference GitHub or other sharing service with some availability of FRET data and simulation models.

CC

Reviewer #3:

Remarks to the Author:

In this paper, Holmstrom et. al probe the nucleocapsid domain (NCD) of the hepatitis C virus core protein for its ability to fold the nucleic acids. They use FRET experiments to first establish that the NCD indeed assists a DNA and an RNA hairpin to fold. Based on decrease in the binding affinity between the hairpin and NCD in "high" salt, they establish that the interactions are driven by charge-charge contacts. Furthermore, they use site-specific labelling to probe the structure of the NCD-nucleic acid complexes, and show that that overall, both the NCD and hairpin become "more" compact when they interact. And finally, they demonstrate that the effect of nanomolar amount of NCD on nucleic acid folding is comparable to that of extremely high (1M) salt concentration NaCl. As more interest is growing in IDPs and their interactions with RNA, this very thorough and timely paper is welcomed. Following are some concerns that should be addressed prior to acceptance.

General Comments:

One suggestion is to provide more background on what this paper is going to be about wrt kinetics and single molecule studies. This is totally unclear from the Abstract and only partially clear in the current Introduction, where the authors simply mention this in passing. A better setup is needed.

It isn't clear that the effects of NCD are chaperoning at all. For instance, Figure 2F doesn't show chaperone behavior per se. Specifically, the equilibrium between F and U for the DNA (or RNA) isn't changed in the absence of the chaperone, like for chaperoning of a protein where the chaperone releases a protein and it stays folded. I don't feel this point, which is important, diminishes the work in any way since what is described is a real thing, but clarity is needed. To me, this is like the use of the word enzyme for ribozyme. Most ribozymes are not enzymes in that they are changed in the reaction and are single turnover; that doesn't make ribozymes less important. But, semantics are important, especially since most of the community uses chaperone as I say above. Clarity is needed if the impact of this paper is to be the highest possible.

Major Comments:

1. My major concern is that many of the IDRs tend to interact with RNA and not with DNA. Yet majority of the experiments in the study has been done with DNA hairpin. Even in the experiments shown in figure S3, there is a four-fold difference in the K_d between the DNA and the RNA hairpin, suggesting some "structural" specificity. As such, could authors provide some guidance on how their findings should be interpreted in the context of what happens in vivo where these IDR-containing proteins are interacting primarily with RNAs?
2. Which brings me to my next concern. As demonstrated in figure 1c, about 20-fold decrease in binding affinity is observed with the addition of 80 mM NaCl. Suggesting that these interactions may be quite sensitive to in vivo salt concentrations which can contain upwards of 140 mM K^+ and both free and weakly chelated Mg^{2+} which can further destabilize these interactions. Which again raises concerns as to whether these interactions really happen in vivo.
3. It appears that 20 out of 25 "positive" charges in the NCD come from arginine residues. Which are known to form bidentate interactions with guanosine. Interestingly, the folding incompetent mutant that authors chose have guanosines at both 5' and 3' end. As such, the binding observed in the "folding incompetent" mutant may be attributed to specific Arg-Guanosine interactions (Luscombe et. al. *Nucleic Acids Res.* 2001 Jul 1; 29(13): 2860–2874). The generality of the results would have been more solid if they had shown similar effects with a different RNA/DNA sequences.

Minor comments

1. Figure 1.
 - a. It is confusing that "DNA Hairpin" appears under the cyan protein.
 - b. The charge on the DNA is not -60. Manning theory says that there is only about -0.1 charges

per phosphate in the present of salt.

c. Panel C. The values on the x-axis appear to be left-shifted for 1.0, 10., and 100.

2. Page 2 line 12, the use of "catalyze" could be interpreted as "chemical reaction".

3. Information in Supplementary figure S2 is could be better presented in the main text by breaking down the panels.

4. p3. Second to last paragraph. "...yielded consistent rate constants (Figures 2c and 2d)" is confusing. I took it as consistent between the panels, which of course is not true. The authors should rewrite to remove this confusing possible interpretation of the current wording.

5. Could authors explain the lack of "more compact" unfolded state (around $E=0.5$) of the hairpin which is seen in figure 1b (cyan) but is absent in figure 4a with NaCl.

Response to reviewers' comments on Manuscript NCOMMS-19-03195

"Disordered RNA chaperones can enhance nucleic acid folding via local charge screening"

We thank the reviewers for their thoughtful comments and constructive criticism, which have helped us further improve the manuscript. A detailed response is given below.

Reviewer # 1:

Holmstrom et al. present an interesting analysis of how intrinsically disordered proteins can act as nucleic acid folding chaperones. This question is particularly interesting given that IDPs or IDRs are present widely in nucleic acid binding and processing proteins. Furthermore, the HCV system is relevant for biology and disease. The authors carefully carry out a range of single molecule fluorescence experiments and find that the highly charged NCD domain of HCV protein increases the efficiency of folding of nucleic acid stem-loop structures, as well as generic compaction. Furthermore, the NCD itself is compacted yet disordered in the complex, reminiscent of IDP-protein complexes studied by the same lab and others in the field. The authors also carry out coarse-grained MD simulations that recapitulate key results and support the electrostatic model. The work is very well done, using a range of experiments and labeled constructs, and new and interesting information about an important biological problem is revealed from the data and simulations. Thus, the work will be of high interest for the readers of this journal.

We would like to thank the reviewer for the positive and encouraging feedback, which has helped us to improve the overall quality and clarity of our manuscript.

I only have a couple of minor comments that the authors can address to improve the manuscript.

As the authors mention early on, RNA folding has been shown to often result in trapped/misfolded intermediates. I would think that the stabilization of compact states shown here would often increase the barrier from these trapped states to native structures. Is this correct? If so, can the authors comment on how the systems might get around this problem?

The reviewer is right in that the mechanism identified in our manuscript only enhances formation of compact conformations and has no way of preferentially promoting native over non-native contacts. We have tried to more clearly capture this point in: the abstract (ln 22); the introduction (ln 37), where the term "functional" may have been misleading; and in the "RNA chaperones as macromolecular counter ions" section (ln 219) by referring to the formation of compact structures rather than the formation of native or functional conformations.

Regarding barrier heights: our results show that chaperone binding stabilizes compact structures via an increase in the folding rate coefficient (i.e., a decrease the height of the folding barrier). Thus, as shown in the simplified free energy diagram below on the left, we thus expect no change in the height of the barrier between folded and misfolded states, in contrast to a scenario where folded or misfolded states are stabilized preferentially (diagram on the right).

As far as I can see, the results here are on DNA folding. Authors, please comment more on the expected differences and potentially more complex behavior for RNA folding.

We indeed focus on DNA in the manuscript, but we also present a data set of experiments performed on an RNA-based hairpin in the supplemental data with results that are very similar to those from the DNA hairpin (Supplementary Figure S2) and suggest that the mechanism we observe is general (at least for RNA and DNA hairpin formation). Given the purely electrostatic character of the proposed mechanism, no fundamental differences between DNA and RNA are expected. The only difference that may arise in RNA folding could be that RNA (because of the 2' OH) is more likely than DNA to have more competing compact conformations.

We have tried to more clearly capture this point in the “*NCD promotes nucleic-acid folding via electrostatic interactions*” section (ln 93-96) and the “*RNA chaperones as macromolecular counter ions*” section (ln 235), in the “*Conclusions*” section (ln 253 and 274).

Reviewer #2:

Holmstrom et al demonstrate how an intrinsically disordered protein can function as an RNA chaperon. By forming only unspecific eletrostatic interaction the chaperon efficiently screens the RNA charges smoothing the free energy landscape and increasing the folding speed and folded state population.

The mechanism is shown to be equivalent to the addition of 1M of NaCl.

A simple physical model is used to simulate the protein-RNA interaction well capturing the observed experimental observation and further demonstrating that the molecular recognition is unspecific and that the resulting effect on the RNA is that of decreasing its average size. The work is extremely well done and definitely worth publishing.

We would like to thank the reviewer for the positive and encouraging feedback.

The only minor observation I feel to add is that

- GROMACS and PLUMED should be referenced

Thanks for the suggestion, both are now referenced in the manuscript (ln 643 and ln 675).

- It would be nice to have a reference GitHub or other sharing service with some availability of FRET data and simulation models.

In accordance with the journal's policies, we have now included a reference to a "Data Availability" section of the manuscript (ln 700-704) where "Source Data" can be found. Details on the simulation model are now included with the re-submission material with web links in the manuscript (ln 644 and ln 676) for documentation on the software.

Reviewer #3:

In this paper, Holmstrom et. al probe the nucleocapsid domain (NCD) of the hepatitis C virus core protein for its ability to fold the nucleic acids. They use FRET experiments to first establish that the NCD indeed assists a DNA and an RNA hairpin to fold. Based on decrease in the binding affinity between the hairpin and NCD in “high” salt, they establish that the interactions are driven by charge-charge contacts. Furthermore, they use site-specific labelling to probe the structure of the NCD-nucleic acid complexes, and show that that overall, both the NCD and hairpin become “more” compact when they interact. And finally, they demonstrate that the effect of nanomolar amount of NCD on nucleic acid folding is comparable to that of extremely high (1M) salt concentration NaCl. As more interest is growing in IDPs and their interactions with RNA, this very thorough and timely paper is welcomed.

We are delighted and encouraged to see that the reviewers consider our report to be thorough and particularly relevant for the field.

Following are some concerns that should be addressed prior to acceptance.

General Comments:

One suggestion is to provide more background on what this paper is going to be about wrt kinetics and single molecule studies. This is totally unclear from the Abstract and only partially clear in the current Introduction, where the authors simply mention this in passing. A better setup is needed.

Thanks for bringing this issue to our attention. We have now adjusted the abstract (ln 20-21 and 25) and introduction (ln 48 and 54) to clarify the methods and approach used in the study.

It isn't clear that the effects of NCD are chaperoning at all. For instance, Figure 2F doesn't show chaperone behavior per se. Specifically, the equilibrium between F and U for the DNA (or RNA) isn't changed in the absence of the chaperone, like for chaperoning of a protein where the chaperone releases a protein and it stays folded. I don't feel this point, which is important, diminishes the work in any way since what is described is a real thing, but clarity is needed. To me, this is like the use of the word enzyme for ribozyme. Most ribozymes are not enzymes in that they are changed in the reaction and are single turnover; that doesn't make ribozymes less important. But, semantics are important, especially since most of the community uses chaperone as I say above. Clarity is needed if the impact of this paper is to be the highest possible.

We agree that correct terminology is important to prevent possible confusion, and we debated this aspect at length during the preparation of the manuscript. In what is generally accepted to be the first presentation of the so-called “RNA chaperone hypothesis” (Herschlag, *JBC*, 1995), Herschlag states:

“‘RNA chaperone’ refers to proteins that aid in RNA folding and is not meant to refer to chaperones made of RNA.”

In our manuscript, we use this definition of “RNA chaperone” in a broad sense. We are aware of more restrictive definitions, which essentially follow Herschlag's suggestion in the same paper:

“RNA chaperones are defined as proteins that aid in the process of RNA folding by preventing misfolding or by resolving misfolded species. This is in contrast to proteins that help ... RNA folding by catalyzing steps along the folding pathway...”

As the reviewer astutely noted, the mechanism we describe for NCD does not specifically prevent or resolve misfolding, and therefore NCD does not fall into this more restrictive category of RNA chaperones. Our decision regarding the use of “chaperone” was reinforced by the common reference to NCD as an RNA chaperone in the existing literature (e.g., Cristofari, *NAR*, 2004; Ivanyi-Nagy, *NAR*, 2006; Ivanyi-Nagy, *NAR*, 2008; Zúñiga, *Virus Research*, 2009; Sharma, *NAR*, 2010; Sharma, *NAR*, 2011; Romero-Lopez, *Scientific Reports*, 2017) and by discussions with Dan Herschlag. Notably, our results include the common “chaperoning assay” for our NCD variant (Supplementary Figure S1). However, following the reviewer’s helpful suggestion, we now clarify this issue in first paragraph (ln 62-66) and at the end of the “Chaperone binding accelerates folding” section (ln 141-144) to alert the reader of possible discrepancies in terminology.

Major Comments:

1. My major concern is that many of the IDRs tend to interact with RNA and not with DNA. Yet majority of the experiments in the study has been done with DNA hairpin. Even in the experiments shown in figure S3, there is a four-fold difference in the K_d between the DNA and the RNA hairpin, suggesting some “structural” specificity. As such, could authors provide some guidance on how their findings should be interpreted in the context of what happens *in vivo* where these IDR-containing proteins are interacting primarily with RNAs?

The detailed differences between DNA and RNA are of course an interesting topic that will be the subject of future investigations. The focus of the present work was to establish a generic picture of charge screening by positively charged IDPs. Correspondingly, we consider the four-fold difference in $^{app}K_d$ between DNA and RNA to be rather small compared to the 30-fold difference in K_d between the folded and unfolded nucleic acids. Our simulations indicate that these changes in affinity can simply be attributed to electrostatics and changes in local charge density rather than evoking additional structure-specific interactions. Nevertheless, the charge densities of nucleic acids do clearly depend on their structures, so from that point of view there may well be some “structural” specificity. In the end, we decided to avoid that terminology for fear that it might misguide the reader into thinking about more classical aspects of structural specificity in protein nucleic acid interactions (like major groove width), which are not required to explain our findings.

We note that the apparent equilibrium dissociation constant is lower for RNA than DNA, indicating that these types of interactions are actually more likely to occur with RNA rather than DNA, which may reflect the greater linear charge density in dsRNA than in dsDNA (Pabit, *NAR*, 2009). Regardless of any small differences in affinity, NCD is a potent chaperone for both RNA and DNA, as indicated by its ability to promote hairpin formation in both cases. Therefore, *in vivo*, NCD would still facilitate folding via the mechanism we describe (i.e., the chaperone will bind to the RNA, compact the unfolded RNA, and increase in the folding rate constant). We have tried to more clearly capture this point in the “NCD promotes folding via electrostatic interactions” section (ln 93-96) and the “RNA chaperones as macromolecular counter ions” section (ln 235).

Finally, we would like to point out that there are also important interactions between charged IDPs and DNA, for instance in the context of the nucleosome and various transcription factors, many of which

contain large positively charged IDRs. Charge screening effects similar to the ones we describe here are likely to contribute to the mechanisms of those proteins, as mentioned in the concluding discussion (In 261-263).

2. Which brings me to my next concern. As demonstrated in figure 1c, about 20-fold decrease in binding affinity is observed with the addition of 80 mM NaCl. Suggesting that these interactions may be quite sensitive to in vivo salt concentrations which can contain upwards of 140 mM K⁺ and both free and weakly chelated Mg²⁺ which can further destabilize these interactions. Which again raises concerns as to whether these interactions really happen in vivo.

The relevance of the salt concentrations we use in our experiments for the cellular situation is an important point that we may not have stressed sufficiently in the manuscript. For the total ion concentrations, one must also consider ions associated with the buffer. The sodium phosphate buffer we use has a concentration of sodium ions of ~75 mM. Together with the NaCl (80 mM), the total monovalent ion concentration (75 mM + 80 mM = 150 mM) is very near physiological monovalent concentrations. In the revised manuscript, we now state explicitly that we use near-physiological monovalent ion concentrations (In 74). Additionally, we changed the horizontal axis of Figure 1d to include contributions from the buffer.

Regarding Mg²⁺, we had not systematically explored the effects of divalent cations and focused on the mechanistic aspects of chaperone activity. However, we now include additional data (Supplementary Figure S4a) demonstrating that inclusion of 1 mM MgCl₂ does not alter the transfer efficiency histogram, indicating that the structural (e.g., transfer efficiencies) and functional (e.g., fraction of bound molecules) properties of NCD are largely insensitive to the concentrations of MgCl₂ in the physiological range. We now also mention this point in the manuscript (In 88-90).

3. It appears that 20 out of 25 “positive” charges in the NCD come from arginine residues. Which are known to form bidentate interactions with guanosine. Interestingly, the folding incompetent mutant that authors chose have guanosines at both 5’ and 3’ end. As such, the binding observed in the “folding incompetent” mutant may be attributed to specific Arg-Guanosine interactions (Luscombe et al. Nucleic Acids Res. 2001 Jul 1; 29(13): 2860–2874). The generality of the results would have been more solid if they had shown similar effects with a different RNA/DNA sequences.

We agree that this is an interesting direction of research and intend to further explore in future work how the protein and nucleic acid sequences modulate the effect we observe by including a number of biologically-relevant structured nucleic acids (and mutants thereof). However, to keep the manuscript focused, we felt that including a DNA hairpin, a folding-incompetent DNA hairpin, and an RNA hairpin would suffice to demonstrate some degree of generality without diluting out the mechanistic findings of the article.

In the experimental design, we concluded that the most conservative folding-incompetent sequence was the mutant where each of the seven nucleotides on the 5’-end of the hairpin was mutated to its complement, eliminating all the Watson-Crick base pairs associated with the original folding-competent hairpin. In both nucleic acids (folding-competent and -incompetent), the vast majority of nucleotides are A, and a change in G content from 5% (3/60) to 10% (6/60) is not expected to substantially alter the binding affinity. In fact, we present experimental evidence to support this claim in the manuscript: We observe a 70 nM K_d between NCD and the incompetent hairpin. We cannot compare this value to the apparent K_d of the folding-competent hairpin (18 nM) because this value arises from a system of

coupled equilibria (binding equilibrium and folding equilibrium), but we can compare the K_d associated with the binding of NCD and the UNFOLDED hairpin. This value can be determined from the HMM analysis of surface trajectories of the folding-competent hairpin (i.e., $6 \text{ s}^{-1} / 8 \cdot 10^7 \text{ M}^{-1} \text{ s}^{-1} = 75 \text{ nM}$, see Fig. 2f). The similarity of these two values indicates that small changes in the G content do not substantially alter NCD's affinity to DNA.

In the revised manuscript, we mention the similarity of NCD affinities to the two sequences more explicitly (ln 137) and include the investigation of alternative sequences as an interesting topic of future research (ln 238).

Minor comments

1. Figure 1.

a. It is confusing that “DNA Hairpin” appears under the cyan protein.

The Figure has been adjusted accordingly.

b. The charge on the DNA is not -60. Manning theory says that there is only about -0.1 charges per phosphate in the present of salt.

The effective charge of polyelectrolytes in solution is indeed reduced due to counter ion condensation and charge regulation and can be estimated using Manning theory and related approaches. In fact, we have recently determined the effective charge of ssDNA using single-molecule electrometry (Ruggeri et al., *Nat. Nanotech.* 2017) and found the effective charge to be approximately half of the structural charge (i.e. the nominal charge based on the sequence), in agreement with Poisson-Boltzmann-type calculations. However, owing to the pronounced surface interactions of positively charged proteins with the nanostructured devices used in single-molecule electrometry, comparable measurements for NCD have not been possible so far. For consistency, we therefore decided to only quote the structural charge. To more clearly make this point, we now use the nomenclature of Ruggeri et al. and refer to “-60” as the “structural charge” (ln 83 and Figure 1a) and included a reference to Ruggeri et al.

c. Panel C. The values on the x-axis appear to be left-shifted for 1.0, 10., and 100.

Thank you for noticing this mismatch. The figure has now been adjusted accordingly.

2. Page 2 line 12, the use of “catalyze” could be interpreted as “chemical reaction”.

We considered the term “catalyze” appropriate since NCD helps the hairpin fold by lowering the effective barrier between U and F, but to avoid any unintentional misunderstanding, we have replaced “catalyze” with “facilitate” (ln 39).

3. Information in Supplementary figure S2 is could be better presented in the main text by breaking down the panels.

We now realize that the first reference to Figure S2 in the main text was potentially confusing. We now refer to Figure S2 in the main text when we specifically mention the salt-dependent measurements (ln 222-223), which should make the figure more intuitive and should not require a panel-by-panel breakdown.

4. p3. Second to last paragraph. “...yielded consistent rate constants (Figures 2c and 2d)” is confusing. I took it as consistent between the panels, which of course is not true. The authors should rewrite to remove this confusing possible interpretation of the current wording.

We now explicitly provide call-outs to the two figure panels separately and rephrased the sentence to prevent a possible misunderstanding (ln 119-121).

5. Could authors explain the lack of “more compact” unfolded state (around $E=0.5$) of the hairpin which is seen in figure 1b (cyan) but is absent in figure 4a with NaCl.

The absence of the “more compact” unfolded state at $E = 0.5$ in Figure 4a is because the figure panel only shows data up to a final concentration of 1 M NaCl, where the transfer efficiency of the “more compact” unfolded state has only increased to a value of $E \approx 0.2$. At higher salt concentrations, it becomes difficult to visualize the mean transfer efficiency of the “more compact” unfolded state because this population is so sparsely populated (since NaCl stabilizes the folded population). As shown in Figure 4c, in order to observe a transfer efficiency of the “more compact” unfolded state that is comparable with the chaperone bound unfolded hairpin, we would need to use a solution with > 4 M NaCl.

Philip Bevilacqua and Raghav Poudyal

Thank you for the rigorous and insightful comments as well as a fully transparent review.

Reviewers' Comments:

Reviewer #3:

Remarks to the Author:

The authors have addressed our concerns. Thank you.